# AI-embodied multi-modal flexible electronic robots with programmable sensing, actuating and self-learning

Junfeng Li [1], Zhangyu Xu[2], Nanpei Li[1], Kaijun Zhang [3], Guangyong Xiong[1], Minjie Sun[1], Chao Hou[2], Jingjing Ji[2], Fan Zhang [2] ✉, Junwen Zhong [3] ✉ & YongAn Huang [2] ✉

Achieving robust environmental interaction in small-scale soft robotics remains challenging due to limitations in terrain adaptability, real-time perception, and autonomous decision-making. Here, we introduce Flexible Electronic Robots constructed from programmable flexible electronic components and setae modules. The integrated platform combines multimodal sensing/actuation with embedded computing, enabling adaptive operation in diverse environments. Applying modular design principles to configure structural topologies, actuation sequences, and circuit layouts, these robots achieve multimodal locomotion, including vertical surface traversal, directional control, and obstacle navigation. The system implements proprioception (shape and attitude) and exteroception (vision, temperature, humidity, proximity and pathway shape recognition) under dynamic conditions. Onboard computational units enable autonomous behaviors like hazard evasion and thermal gradient tracking through adaptive decision-making, supported by embodied artificial intelligence. In this work, we establish a framework for creating small-scale soft robots with enhanced environmental intelligence through tightly integrated sensing, actuation, and decision-making architectures.

High mobility and embodied intelligence represent foundational requirements for achieving autonomous functionality in small-scale soft robots, particularly for applications requiring adaptive interactions with unstructured environments. Advances in soft robotics have yielded diverse locomotion strategies, including crawling[1–3], hopping[4–6], and rolling[7–9] systems that balance operational safety with environmental compliance. Challenges persist in enabling these robots to execute complex missions such as autonomous search-and-rescue operations or dynamic terrain navigation[10–12]. While biological counterparts like insects demonstrate adaptability through integrated sensing and motion control, most artificial systems remain constrained to predefined locomotion patterns optimized for specific scenarios. Although some soft robots leverage machine learning algorithms to recognize terrains or navigate obstacles[13], responding to dynamic environments continues to pose challenges.

The emerging framework of embodied artificial intelligence (AI) presents potential for addressing these limitations by unifying sensory perception, decision-making, and actuation within cohesive robotic architectures[14–17]. Inspired by arthropods such as millipedes—organisms that achieve environmental adaptation through malleable bodies, sensory acuity, and innate behavioral strategies[18]—our approach reimagines robotic design through the lens of bioinspired modularity (Fig. 1a). Traditional soft robots often face fundamental trade-offs between structural compliance and functional complexity, particularly when integrating sensing, computation, and actuation subsystems[19]. Our solution leverages advancements in flexible electronics and

[1]School of Mechanical and Electronic Engineering, Wuhan University of Technology, Wuhan, China. [2]State Key Laboratory of Intelligent Manufacturing Equipment and Technology, Huazhong University of Science and Technology, Wuhan, China. [3]Department of Electromechanical Engineering, Centre for Artificial Intelligence and Robotics, University of Macau, Macau, China. ✉e-mail: fanzhang@hust.edu.cn; junwenzhong@um.edu.mo; yahuang@hust.edu.cn

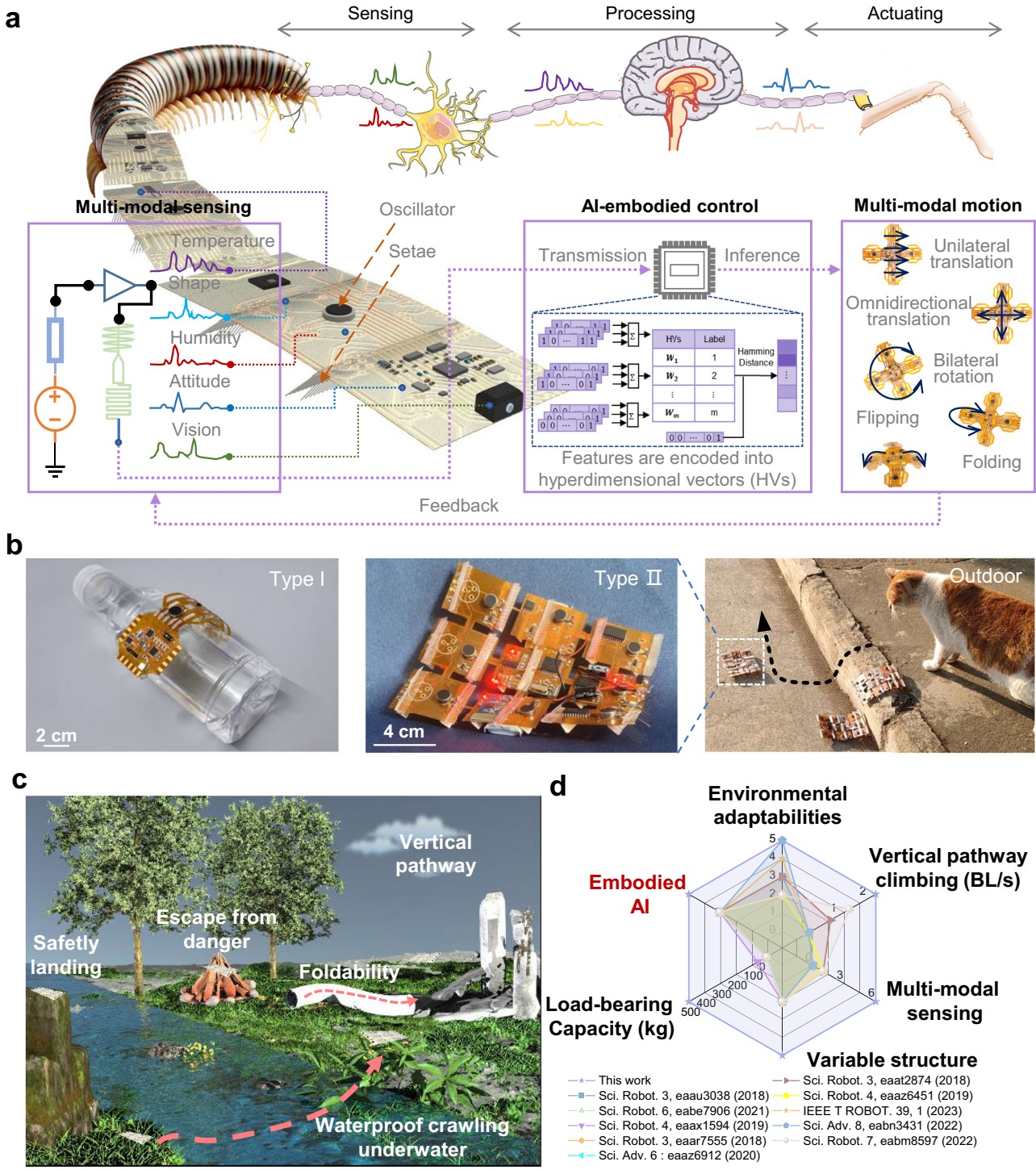

**Fig. 1 | Millipede–inspired AI-embodied FEbots. a** Schematic diagram illustrating a millipede-inspired FEbot capable of multi-modal sensing, motion, and onboard hyperdimensional computing through functional mimicry of a central nervous system. **b** Photo of a millipede-like FEbot (Type I) with high flexibility and a square-shaped FEbot (Type II) demonstrating outdoor operation. **c** Schematic diagram illustrating FEbots interacting with the physical world via autonomous sensing-knowing-actuation. **d** Performance comparison between FEbots and reported small-scale soft robots.

oscillatory actuation to create integrated systems where structural adaptability coexists with embedded computational capacity.

Key to this development are flexible electronics robots (FEbots) that combine modular architectures with oscillatory actuation mechanisms. Unlike robots requiring individual leg trajectory control[3,20], FEbots use distributed setae arrays to simplify actuation while maintaining environmental adaptability[21–25]. Their locomotion employs oscillatory actuation driven by periodic bending of super-

elastic alloy (SSMA) setae. SSMA provides elastic deformation capacity, corrosion resistance, and durability (Supplementary Table 1), enabling integration with flexible electronics.

Integrated flexible electronics enable environmental perception through embedded sensors (vision, proximity, temperature, humidity) and processing units implementing hyperdimensional computing algorithms. Modular reconfiguration—including adjustable setae densities and electronics layouts—allows transitions between specialized

configurations: a Type I design emphasizes contortion capability for confined spaces, while a square-shaped Type II configuration enhances stability for outdoor navigation (Fig. 1b).

We demonstrate FEbots achieving environmental interaction through coordinated actuation and embedded AI. The system accomplishes multi-modal locomotion, including vertical pathway climbing, confined space traversal, and omnidirectional movement. Sensor fusion enables terrain classification and obstacle avoidance through integrated sensory systems, with decision-making processes handled by onboard computing chips. This integration of flexible electronics, distributed actuation, and embodied AI provides a framework for developing autonomous soft robots capable of operating in dynamic environments, as illustrated in Fig. 1c, d, and Supplementary Table 2[3,10,11,26–32].

## Results

### Analysis of sliding locomotion principles and key parameter optimization

The FEbot demonstrates exceptional multi-terrain mobility through bioinspired setae arrays and adaptive body segmentation. Unlike previous research mainly considering the forward movement in the moving direction, we present a modeling method for predicting the forward-backward sliding displacement in X direction, and vibration in Y direction. To elucidate the oscillation-driven locomotion mechanism of its individual units, we analyze four critical factors: input forces, sliding dynamics, anisotropic friction, and geometric parameters. When an oscillator operates, it generates circumferential forces in the horizontal plane and vertical plane, which results in a two-dimensional movement (2D) of unit. The main motion arises from asymmetric friction modulation during three distinct states in one actuation cycle, driven by periodic horizontal ($F_x$) along X direction and vertical ($F_y$) along Y direction (Fig. 2a). In state-I (compression phase, $t_1$), the unit reaches its lowest position ($\Delta s_l$) with setae fully compressed; in state-II (backward sliding, time from $t_1$ to $t_2$), during vertical ascent to the highest position ($\Delta s_h$), backward sliding displacement ($\Delta s_b$) occurs as $F_x$ exceeds the backward friction resistance ($F_{f-}$); in state-III (forward sliding, time from $t_2$ to $t_3$), during descent to $\Delta s_l$, forward displacement ($\Delta s_f$) is achieved as $F_x$ overcomes the reduced forward friction resistance ($F_{f+}$).

The net displacement per cycle is:

$$\Delta s_1 = \Delta s_f - \Delta s_b \tag{1}$$

Anisotropic friction originates from backward-oriented setae (e.g., earthworm)[22–25,33], where angled setae tips generate direction-dependent resistance ($F_{f-} > F_{f+}$). For sustained forward locomotion, the force hierarchy must satisfy:

$$|F_x| \gg |F_{f-}| > F_{f+} \tag{2}$$

To make quantitative predictions of the locomotion gait, we establish the dynamic model by employing "Cosserat elastic rod theory" for setae deformation, while combining spring-damper theory to analyze the reciprocal sliding behavior of the robot in the X-direction. This approach validates the accuracy of the FEbot's dynamic model. Simulations incorporate measured oscillator forces and geometric parameters (Supplementary Table 3), with numerical methods detailed in Supplementary Notes 1, 2 and visualized in Supplementary Figs. 1–6.

A prototype 2.2 × 1.5 cm unit (0.83 g) demonstrates locomotion at a speed of 46.8 mm/s (2.13 BL/s) under a 3 V driving voltage (260 Hz). High-speed video (3000 frames per second) captures the motion of the unit on a smooth aluminum substrate during steady-state operation, with nonmaximal running performance observed (Supplementary Movie 1). As shown in Fig. 2b, the mass center (point $C_G$) follows a spiral-like trajectory, and simulation results (Supplementary code 1)

align with experimental motion when incorporating the input force from Supplementary Fig. 4a. The unit exhibits reciprocating motion in the X-direction (backward displacement: $\Delta s_b = 0.085$ mm at 1.33 ms; forward displacement: $\Delta s_f = 0.265$ mm at 4 ms, cycle time $T_U =$ ms, Fig. 2c) and vertical oscillation in the Y-direction ($\Delta s_2 = \Delta s_h$-$\Delta s_l$, highest position: $\Delta s_h = 4.58$ mm at 0 ms; lowest position: $\Delta s_l = 4.46$ mm at 2 ms, Fig. 2d). Positional data in the X-direction over five cycles are shown in Supplementary Fig. 6b.

When the driving voltage exceeds 3 V (e.g., 4 V, 400 Hz), the unit transitions to unstable crawling with random hopping. Structural refinement of setae geometry enhances motion stability. Key geometric parameters include seta length ($L$), diameter ($d$), and contact angle ($\theta$) between the seta and substrate (Fig. 2a). A larger prototype (5 × 3 cm, 1.66 g, Supplementary Fig. 7a) is tested on paper substrates to evaluate parameter-dependent performance. For a unit with $L = 5$ mm, $d = 0.1$ mm, and $\theta = 45°$, forward speed correlates with frequency, peaking at 75.9 mm/s (1.52 BL/s) under 450 Hz (5 V) (Fig. 2e). Adjusting parameters to $L = 5$ mm, $d = 0.1$ mm, and $\theta = 60°$ yields a speed of 107.3 ± 3.02 mm/s (Fig. 2f, yellow line). Increasing $L$ to 7 mm under identical conditions improves speed to 109.5 ± 3.84 mm/s (Supplementary Movie 2). Units with $d = 0.07$, 0.2, and 0.3 mm ($\theta = 60°$) achieve maximum speeds of 117.3 ± 7.89 mm/s (blue line), 93.6 ± 6.27 mm/s (red line), and 112.8 ± 9.49 mm/s (green line) at seta lengths of 3, 13, and 15 mm, respectively. Speed reductions occur when seta lengths exceed 3, 7, 27, and 21 mm for $d = 0.07$, 0.1, 0.2, and 0.3 mm, corresponding to transitions from point ($F_{f-} > F_{f+}$) to linear ($F_{f-} \approx F_{f+}$) contact between setae and substrate (Supplementary Fig. 7b).

Variations in seta angle ($\theta$) for a unit with $L = 7$ mm and $d = 0.1$ mm alter forward speeds (Fig. 2g). At $\theta = 75°$, the unit reaches a maximum speed of 123.1 mm/s (2.46 BL/s), however motion variability increases compared to $\theta = 60°$. Slope-climbing tests (Supplementary Fig. 8a) show that units with $L = 7$ mm and $\theta = 60°$ ($d = 0.07$, 0.1, 0.2, 0.3 mm) maintain a speed of 14 mm/s on slopes with maximum inclines of 3.6°, 18.0°, 14.4°, and 10.8°, respectively. However, payload tests (Supplementary Fig. 8b) demonstrate that a unit with $d = 0.2$ mm carries 18 g (10.8 times of its mass) at 29.3 mm/s (0.59 BL/s) (Supplementary Movie 3). Therefore, the configuration with $L = 7$ mm, $d = 0.1$ mm, and $\theta = 60°$ exhibits balanced performance in speed, stability, and climbing capability except load capacity.

### Multi-modal locomotion capabilities

Biological systems such as millipedes and snakes exhibit adaptive locomotion strategies to traverse diverse terrains, a trait enabled by evolutionary adaptations[21,34]. This principle has guided the development of soft robots with multi-modal mobility[35,36], particularly those employing flexible actuation. A persistent challenge remains in achieving high terrestrial maneuverability through lightweight, structurally simple actuation systems. To address this, we implement a modular design framework combining flexible electronic modules (control, actuation) with reconfigurable setae modules, enabling rapid prototyping of FEbots.

Four flexible electronic modules are developed (Fig. 3a(i) and Supplementary Fig. 9): actuation module with integrated strain sensor (BF120-10AA), temperature/humidity sensor module (HDC1080), proximity sensor module (QRE1080), and central controller (NRF52832) with inertial measurement unit (MPU6050). The controller features front/back interconnect pads for module integration via conductive tape. Sensor-equipped actuation modules interface with the controller through dual circuits, enabling FEbot reconfiguration through module rearrangement. Setae variants (foldable/non-foldable) permit distinct morphologies, including: millipede-inspired design for narrow pathway navigation (Type I, Fig. 3a(ii, iii)) and square configuration supporting multi-modal motion (Type II, Fig. 3a(iv)).

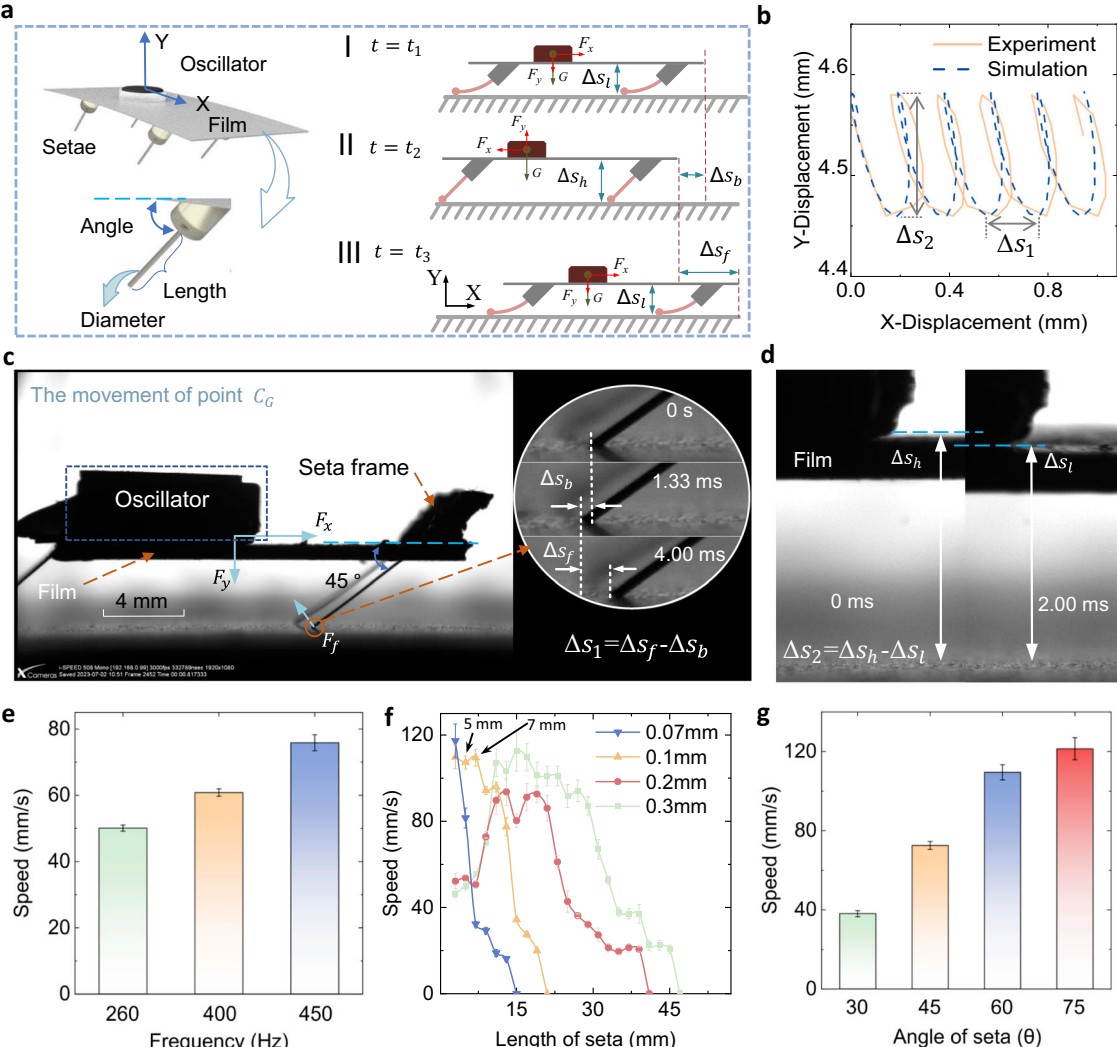

**Fig. 2 | Locomotion principles and geometric parameter optimization. a** Key geometric parameters of the seta and locomotion analysis of a single FEbot unit during one driving cycle. **b** Trajectory of the mass center (derived from high-speed imaging, Supplementary Movie 1) compared to simulation results. **c**, **d** Sequential optical images capturing motion profiles in the X- and Y-directions during one operational cycle, respectively. **e** Forward speed as a function of driving frequency for a unit with $L = 5$ mm, $d = 0.1$ mm, and $\theta = 45°$. **f** Forward speed versus seta length for units with $d = 0.07, 0.1, 0.2, 0.3$ mm and $\theta = 60°$. **g** Forward speed as a function of seta angle ($\theta$) for units with $L = 7$ mm and $d = 0.1$ mm.

For example, a millipede-like FEbot (Type I) with setae modules fixed on both sides of a flexible electronic module with an oscillator that can move forward in confined or even vertical gap without complicate steering mechanism is presented firstly (Supplementary Fig. 10a). With the understanding of structural optimization principle, a $4 \times 2 \times 3$ cm prototype (Type I, $d = 0.1$ mm, $\theta = 60°$, mass 1.75 g, tethered) navigates vertical pathway (2 cm inner width) at 87.6 mm/s (2.19 BL/s) (Fig. 3b, Supplementary Movie 4). The design enables carrying 8.9 g payloads (5.1 times of self-weight) at a velocity of 8.5 mm/s in vertical channels (2 cm inner width). It can also go through a 14 mm-wide passages (70% body width) at a velocity of 25.6 mm/s (Fig. 3c). Horizontal channel tests (Supplementary Fig. 10b) show peak velocity of 123.6 mm/s (3.09 BL/s) in 2 cm-wide paths and 8.6 mm/s (0.22 BL/s) under 15 g loading (8.5 times of self-weight) (Supplementary Fig. 10c, d).

An additional Type I FEbot prototype ($13.2 \times 4.4 \times 1$ cm, 3.9 g, tethered) integrates two foldable setae modules with three flexible electronic modules (one controller, two actuators) (Fig. 3d(i), Supplementary Fig. 11a). Heating SMA springs with 0.2 A current increases the foldable setae angle $\theta$ from 0° (linear contact, $F_{f-} \approx F_{f+}$) to 45°

(point contact, $F_{f-} > F_{f+}$), inducing anisotropic substrate friction (Supplementary Fig. 11b, Supplementary Movie 5). The controller regulates oscillators on interconnected actuation modules via conductive tape, enabling bidirectional motion in confined spaces at 20 mm/s (1.52 BL/s) under 3.7 V actuation (Supplementary Movie 6). The foldable setae design exhibits mechanical resilience of heavy load bearing capacity, sustaining 500 kg car wheel compression ($2.5 \times 10^5$ times of self-weight) without functional degradation (Fig. 3d(ii), Supplementary Movie 7).

The square-configuration FEbot (Type II, Fig. 3e) combines one controller, four actuators, and ten setae modules. Directional control is achieved by setae orientation adjustments: linear motion (forward/backward at 27.93/23.11 mm/s), steering (right/left turns at 33.20/20.24 mm/s) and rotation (clockwise/anticlockwise at 43.93/42.33°/s) (Supplementary Movie 8). The insufficient directional motion control accuracy of FEbots primarily stems from following mechanical characteristic variations: (1) friction coefficient discrepancy: manufacturing tolerances and assembly precision limitations lead to the differences in friction coefficients between setae modules and contact surfaces; (2) directional deviation accumulation: the speed mismatch between

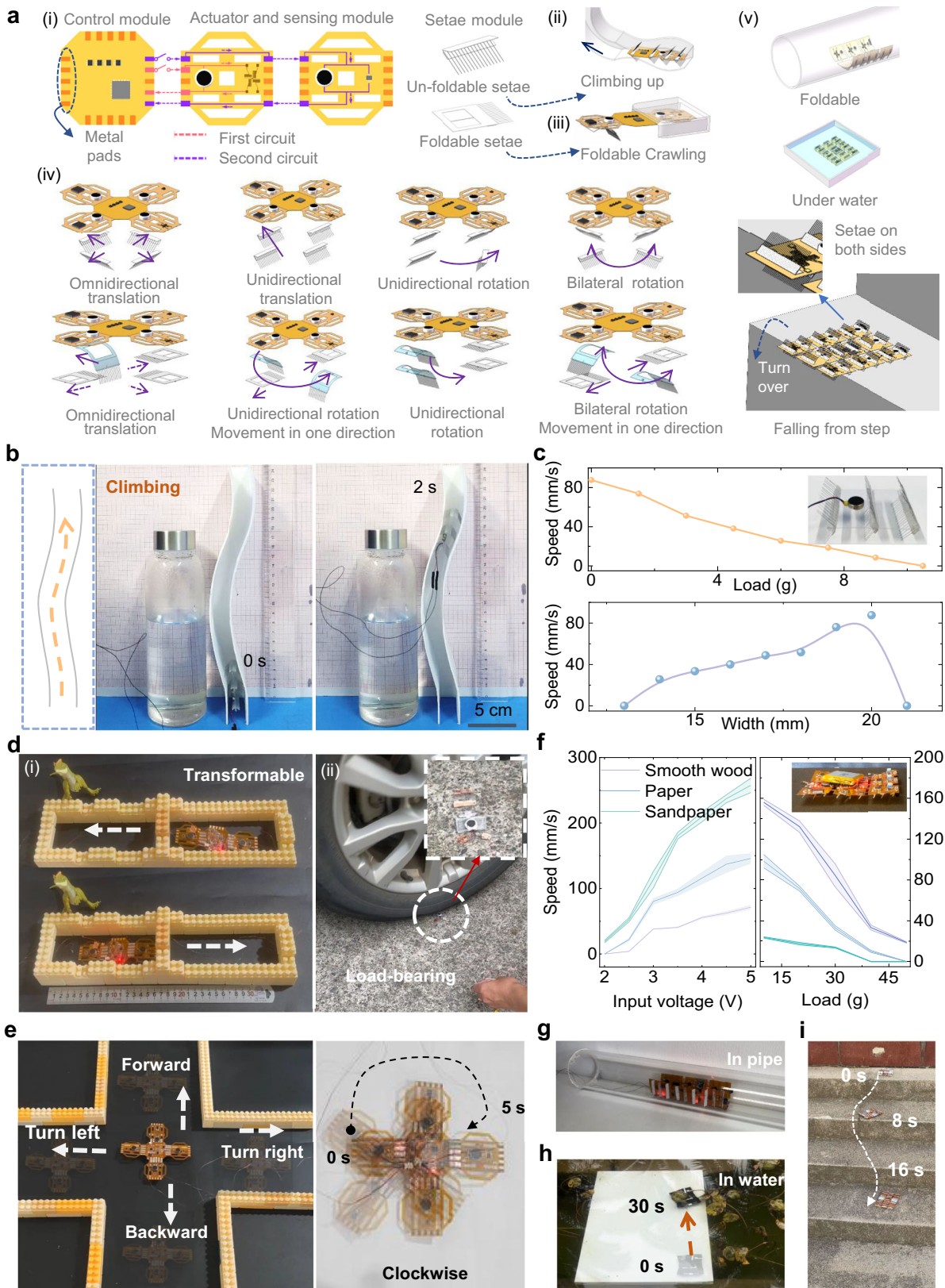

**Fig. 3 | Multi-environment locomotion tests. a** (i) Programmable FEbot assembly design, (ii) Vertical pipe climbing demonstration, (iii) Navigation through confined spaces, (iv) Modular setae arrangements for motion adaptation, (v) Terrain adaptability across substrates. **b** FEbot (Type I) ascending a curved vertical pipe at 87.6 mm/s. **c** Load-speed correlation during vertical climbing (top), width-dependent velocity profiles (bottom), with bilateral setae configuration shown inset. **d** Bidirectional motion in FEbot (Type I) through SMA-driven foldable setae, with integrated load capacity testing. **e** Steering control via setae orientation modulation (Type II). **f** Surface-adaptive performance metrics: velocity and loading capacity on sandpaper, paper, and polished wood (Type II). **g** Pipe navigation using compact folded configuration (Type II). **h** Aquatic locomotion on submerged surfaces (Type II). **i** Stair descent capability (Type II).

symmetrically distributed actuation modules induces cumulative directional errors during motion. To enhance control precision, a sensor-based closed-loop control scheme is proposed in the following sections: Implement optical encoders or attitude sensors to establish a pose perception system, which can achieve closed-loop regulation of turning angles through real-time feedback. This approach enables continuous trajectory correction, effectively suppressing directional drift through electromechanical coupling control. Different types of programmatically assembled FEbots are listed in Supplementary Table 4.

A nine-unit, square-shaped FEbot (Type II) with independent actuation for forward motion and steering operates (Supplementary Fig. 12a). Motion performance varies across terrains: smooth wood (Ra = 0.5 μm), A4 paper (Ra = 2 μm), and sandpaper (Ra = 8 μm) (Fig. 3f). At 5 V, sandpaper achieves the highest linear speed (257 mm/s, 1.98 BL/s) due to increased friction. Speed decreases with added mass, transporting a 50 g load (2.8 times of weight) at 18.7 mm/s. This FEbot with folding shape crawls in a smooth pipe at 17.3 mm/s (0.13 BL/s) (Fig. 3g, Supplementary Movie 9). Integrated photodiodes as optical encoders enable adaptive behavior (Supplementary Fig. 12b). Activating all eight segments drives forward motion, while activating three left or right segments triggers right or left turns, respectively. The FEbot autonomously navigates a 28.5 cm Ω-shaped path in 83 s using onboard photodiode control (Supplementary Movie 10), showcasing closed-loop capability. In addition, the untethered FEbot (Type II) integrated with two batteries (3.7 V, 400 mAh) can be directionally controlled through a Bluetooth-enabled smartphone interface to modify its movement path (Supplementary Fig. 12c). When encapsulated with a waterproof coating, the system demonstrates linear motion at 9 mm/s (0.09 BL/s) on submerged surfaces and executes turning maneuvers with an angular velocity of 3.75°/s (Fig. 3h, Supplementary Movie 11). The reduced crawling speed underwater compared to terrestrial operation aligns with the higher fluid resistance in aquatic environments. A FEbot configuration featuring bilateral setae maintains step-climbing capability following inversion (Fig. 3i, Supplementary Movie 12). The system demonstrates 25 min of continuous operation with 2.5 Wh power consumption.

## Multi-modal sensory perception capabilities

While the FEbots demonstrate mobility and adaptability, integrating environmental perception capabilities is essential for acquiring target information to support effective operation[37,38]. A FEbot (Type I) integrating actuation, control modules, and multiple sensors-including an attitude sensor, thermal/humidity sensor, strain sensor, and miniature camera-is shown in Fig. 4a. Prior work indicates that flexible electronic module technology can incorporate additional sensor types, such as pressure, heat flow, and shear force sensors[39], forming a versatile platform. The FEbot enables monitoring of its body shape and attitude (proprioception) and detection of external stimuli (exteroception) as it traverses an S-shaped pathway (curvature = 0.1 cm⁻¹) at an average speed of 28.6 mm/s (0.21 BL/s) (Fig. 4b, Supplementary Movie 13). The miniature camera provides real-time video capability for environmental monitoring (Fig. 4c, Supplementary Movie 14). Proprioception measurements reveal that the bending posture varies from 0° to 180° during the first 15 s and returns to 0° between 15 and 35 s, with body curvature measured at −0.09 cm⁻¹ (8–18 s) and 0.11 cm⁻¹ (19–35 s). Exteroception data indicate relative humidity levels between 42.8 and 43.8% RH along the pathway, while temperature increases from 33.7 to 39.7 °C during traversal (Fig. 4d).

An 11 × 3 cm FEbot (Type I) incorporating strain sensor arrays for pathway shape recognition is illustrated in Fig. 4e. The diaphragm strain sensor employs a Wheatstone bridge circuit to detect resistance changes induced by bending deformation. Sensor calibration details are provided in Supplementary Note 3 and Supplementary Fig. 13. A reconstruction algorithm based on homogeneous transformation

matrices and D-H parameter methods is developed to estimate pathway geometry using measured curvatures (Supplementary Note 4, Supplementary Figs. 14–16). Figure 4f compares actual and reconstructed pathway shapes with arc radii of 86, 67, and 48 mm, showing strong agreement. The method is further validated by reconstructing a complex S-shaped pathway (Supplementary Fig. 17, Supplementary Movie 15). These results suggest that the approach could enable soft robots to actively detect features in confined spaces.

## FEbot with embodied AI

In nature, insects exhibit instinctive responses to avoid threats, driven by evolutionary pressures. The integration of embodied intelligence into robotic systems enables adaptive behaviors in dynamic and unpredictable environments[18,40,41]. Here, a 4.2 × 8.4 cm untethered FEbot (Type I) is introduced, configured with a controller module incorporating sensors such as a proximity sensor (Supplementary Fig. 18a). Figure 5a illustrates the control strategy based on environmental inputs. To enable autonomous decision-making through on-chip computing, hyperdimensional computing (HDC) is implemented, supported by a self-learning framework that offers rapid learning, low latency, and compact models[42,43]. This approach is particularly suited for in-sensor inference compared to conventional machine learning methods. Intelligent decision-making involves training, inference, and HDC processes. As demonstrated in prior work[44], offline environmental recognition is achieved through HDC, which includes training and inference phases. During training, time-domain features are extracted and encoded into hyperdimensional vectors (HVs) via an HD encoder:

$$H_{HV} = HVE(f_{TD}) \tag{3}$$

where $f_{TD}$ represents time-domain features, $HVE$ denotes the encoding function, and $H_{HV}$ corresponds to the encoded vectors. HVs of the same category are aggregated into class vectors and stored in an associative memory (AM). During inference, test data undergoes similar feature extraction and encoding. The Hamming distance between test HVs and class HVs in the AM determines classification:

$$Ham\left(\mathbf{E}_j, \mathbf{W}_i\right) = \frac{1}{D} \sum_{\varpi=1}^{D} 1_{\mathcal{B}_j(\varpi) \neq W_i(\varpi)} \tag{4}$$

where $\mathbf{E}_j$ is the encoded HV query of the $j$-th test sample, $\mathbf{W}_i$ is the $i$-th categorized HV, and $D$ is the dimension of the HV. Classification relies on identifying the smallest Hamming distance. Recognition accuracy is evaluated by comparing predictions with ground-truth labels.

The HD space is dynamically updated to support real-time learning. On-chip computations drive the recognition system, and a microprogrammed control unit adjusts the pulse-width modulation (PWM) duty cycle to regulate oscillator frequencies:

$$PWM_z = pwm_{set}(Ham), z = 1, 2 \tag{5}$$

where $pwm_{set}$ is the control signal determined by the robot posture according to the Hamming distance, $PWM_z$ is $z$-th oscillator's value of PWM.

For example, the FEbot integrated with AI avoids threats through rapid perception and feedback (Fig. 5b). Under safe conditions, it moves at 4.4 mm/s with a low PWM duty cycle. When a hand approaches, the duty cycle increases, enabling escape at 105 mm/s (Fig. 5c, Supplementary Movie 16). A second FEbot (Type I), consisting of a controller module, two actuator modules equipped with temperature and proximity sensors, and uniformly aligned setae modules, demonstrates steering by adjusting actuator frequencies independently (Supplementary Fig. 18b). Oscillators on either side operate at different frequencies: low-temperature sensing activates oscillator1 for inward tilting, while high-temperature sensing activates oscillator2 for

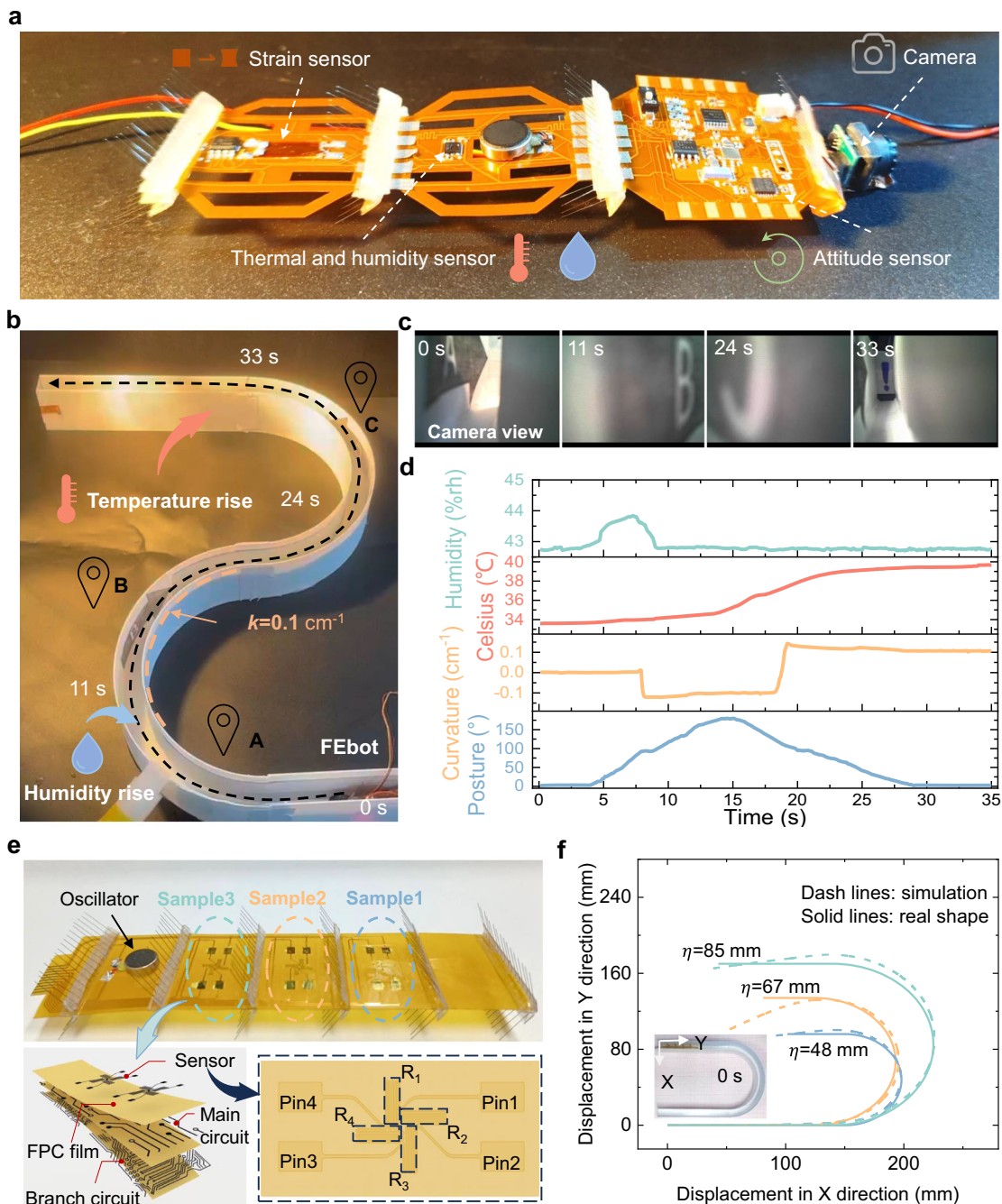

**Fig. 4 | Multi-modal sensing characterization. a** FEbot (Type I) configuration with integrated sensory modules. **b** Experimental setup for simultaneous proprioceptive and exteroceptive measurements. **c** Environmental monitoring using a miniature camera (1 g mass; model MC900, 3RDEE Co., Ltd.). **d** Recorded parameters including body posture, curvature, humidity, and temperature during locomotion. **e** Strain sensor array configuration for pathway morphology detection. **f** Comparison of reconstructed and actual pathway geometries with arc radii of 86, 67, and 48 mm.

outward tilting. This FEbot navigates obstacles in environments with temperature gradients using HDC (Fig. 5d, e, Supplementary Fig. 19, Supplementary Movie 17). Snapshots taken at 6-s intervals show its trajectory. The FEbot also identifies isothermal lines (35 and 45 °C) in thermal fields (Supplementary Fig. 20). These results demonstrate that the AI-integrated FEbot performs autonomous adaptive behaviors, such as multimodal perception, real-time decision-making, and environmental interaction, using onboard resources.

## Discussion

Drawing inspiration from arthropod locomotion and perception principles, we develop modular FEbots integrating flexible electronics and sensory systems. A dynamic model elucidates the oscillation-driven locomotion mechanism of individual units[45], predicting combined sliding (X-axis) and vibrational (Y-axis) motion patterns. The FEbots demonstrate programmable multi-modal mobility, including vertical pathway climbing, omnidirectional navigation, and confined space traversal, complemented by multi-environment operation capabilities in aquatic and structured terrestrial settings. Integrated multimodal sensing encompasses shape recognition, temperature profiling, humidity detection, proximity measurement, attitude monitoring, color discrimination, and visual perception. Performance metrics reveal a payload capacity of 10.8 times of body weight and structural tolerance to $2.5 \times 10^5$ times of self-mass. An embedded AI

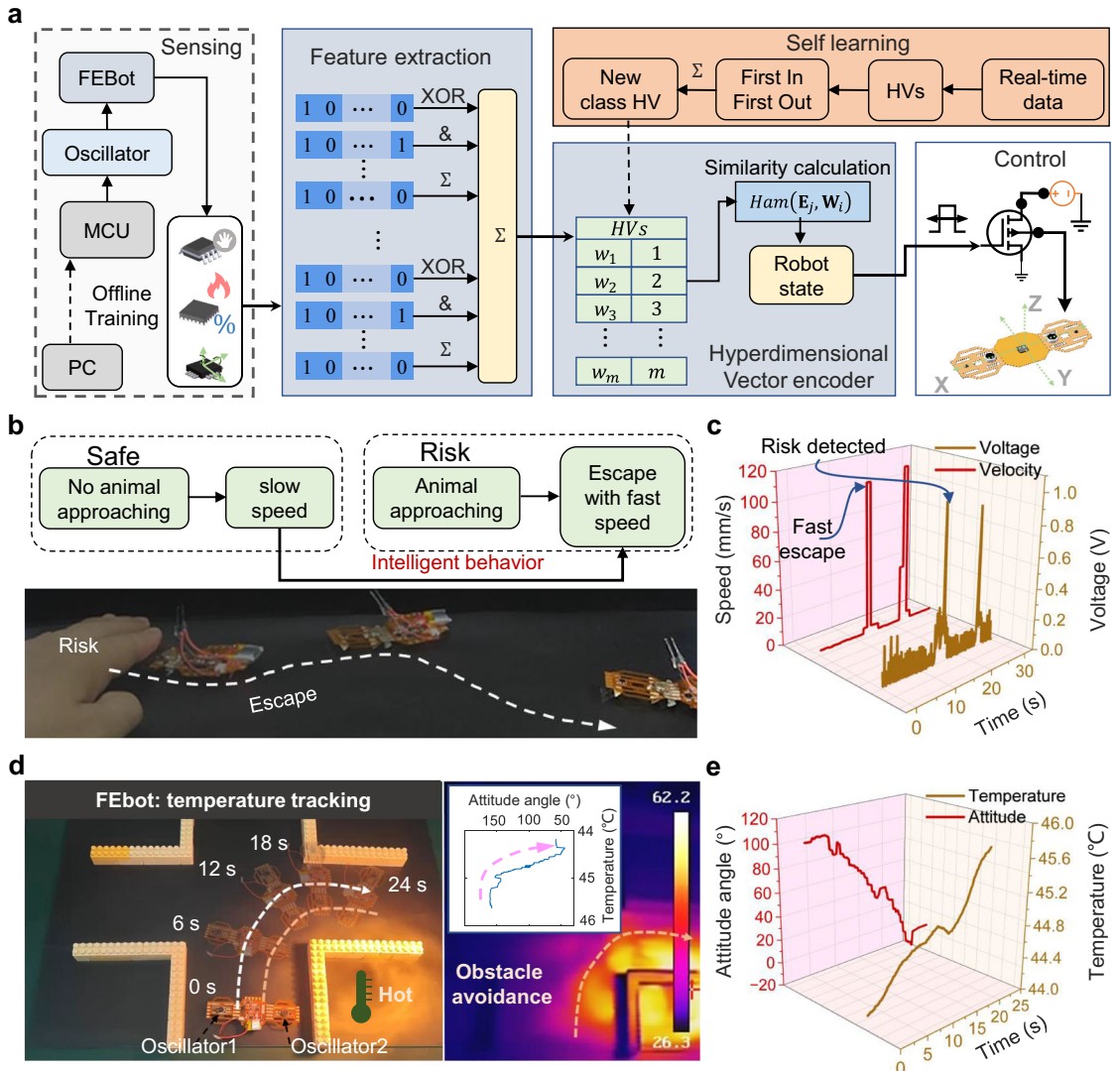

**Fig. 5 | Autonomous behaviors of AI-integrated FEbot. a** Control framework for the FEbot using hyperdimensional computing and self-learning. **b, c** FEbot evading hazardous conditions through real-time perception and feedback. **d** FEbot tracking a temperature zone (47.5 ± 2.5 °C) while avoiding obstacles, illustrated by a photographic sequence of a 90° turn completed in 24 s, with corresponding infrared imaging. **e** Temperature tracking performance and attitude angle measurements.

architecture enables autonomous hazard navigation and thermal field characterization, while modular design facilitates functional reconfiguration for diverse applications.

Current challenges and future research directions focus on: (i) terrain adaptability: while demonstrating robust mobility, the robots exhibit compromised locomotion on soft substrates due to setae sinking into loose surfaces, necessitating biomimetic design enhancements for mobility on loose surfaces; (ii) system scaling: miniaturization strategies and environmental hardening to extend underwater endurance and heterogeneous terrain adaptability; (iii) cognitive efficiency: neuromorphic perception-control systems inspired by insect neurodynamic to optimize autonomous decision-making.

## Methods

### Fabrication of locomotion analysis units
Locomotion units employ a 3D-printed PLA framework (FINDER, www.sz3dp.com). The prototype (Supplementary Fig. 21) measures 15 × 10 × 3 mm (length, width, height). SSMA setae and oscillators integrate with the PLA frame via mechanical fixation. Units in Supplementary Movies 2, 3 use identical fabrication methods with colored PLA.

### Strain sensor fabrication
The process involves five key steps:
(1) Spin-coating PMMA on glass substrates followed by thermal curing (5 min, 95 °C);
(2) Polyimide deposition (220 °C, 4 h for imidization);
(3) Photolithographic patterning (AZ5214 photoresist, 8.5 s exposure);
(4) Platinum sputtering (0.8 Pa, 25 min);
(5) Lift-off completion via acetone/DI water rinsing and thermal tape release. Parallel strain sensors (17 × 6 × 0.01 mm) with 0.09 mm sensing grids form the final structure (Supplementary Fig. 22, Supplementary Table 5).

### Characterization
(1) Oscillator force measurements utilize a four-channel 24-bit DAQ system (AUMANYU DAQ-580i);
(2) High-speed videography (i-SPEED 508 Mono, 3000 fps) captures locomotion kinematics;
(3) Strain sensor calibration combines uniaxial tensile testing (INSTRON-5944) with resistance monitoring (Keithley DAQ6510);

(4) Infrared imaging (HIKVISION DS-2TPH10-3AUF) maps thermal field distributions.

## Data availability

The data that support the findings of this study are available within this article and its Supplementary Information. All data are available from the corresponding author upon request.

## Code availability

The code described here is deposited in Code Ocean (https://doi.org/10.24433/co.1197438.v1.0).

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

## Acknowledgements

This work receives partial support from the National Natural Science Foundation of China (grant nos. 52525502 and 52188102), the China Institute of Atomic Energy (grant no. KFZC2021010301), and the Science and Technology Development Fund, Macau SAR (grant no. 0117/2024/AMJ).

## Author contributions

J.L. conceived and supervised the research, conducted experiments, and drafted the manuscript. G.X., N.L., Z.X., M.S., and Y.H contributed to prototype design, fabrication support, and experimental validation. C.H. developed strain sensor architectures. J.J. performed computational simulations. K.Z. coordinated data visualization. F.Z., J.Z., and Y.H. formulated the conceptual framework, supervised the project, and edited the manuscript.

## Competing interests

The authors declare no competing interests.
