## [Transparent Peer Review file · Nature Communications]

AI-embodied multi-modal flexible electronic robots with programmable sensing, actuating and self-learning

Corresponding Author: Professor Junfeng Li

Version 0:

Reviewer comments:

Reviewer #1

(Remarks to the Author)

This paper presents modular, flexible, centimetre scale robots called FEbots, inspired my millipedes and designed for embodied intelligence control. The use of vibrating SMA setae for multi-modal modular locomotion is of particular interest. Good performance on flat surfaces for this robot and interesting morphability through SMA actuation of setae and adaptation to different locomotion modes is also shown.

The supplementary materials include videos and

The standard of English in the paper is reasonable with good spelling and sentence structure, but still needs improvement. Please go over the paper carefully with a grammar checker or experienced writer of English and correct the occasional English grammatical inconsistencies. Examples of this that can be corrected in the abstract alone include definite articles "The FEbots proposed in this work"; "Inspired by the millipede" prepositions "(e.g. on smooth and rough ground, stairs, underwater, in vertical pipes)", indefinite articles "assisted by a modularization design method", and plural "in complex environments".

Fig. 1c caption 3 should read "Underwater crawling with waterproofing" or "Waterproof crawling underwater". Caption 6 "Climb up in slit with compressibility" should be more clearly worded - should this be "vertical pathway" as in Movie S4?

Please keep consistent terminology across your materials.

Fig. 1d is an important comparison to previous work, but the small size of this figure and the difficulty in discriminating the provided references by overlapping color regions on the radial graph makes it almost unreadable. This should be enlarged and the separate data more clearly differentiated (perhaps line-only with different line styles?) on to provide a clearer fit with Table S1.

You mention that "The anisotropic friction between setae and ground is the key point for (the) locomotion mechanism" (again, please add definite article), but it is not clear how this anisotropic friction arises. Reference 34 mentions "bioinspired microrobotic legs with claws that are capable of inducing anisotropic friction" but it does not appear you are using such a design as movie S1 shows only a SMA wire. Are you relying on a squared-off corner on the setae to produce the difference between F_{f-} and F_{f+} or is it entirely or is it due to the $\theta=60$ angle of movement in which case it is more akin to a ratchet action? The latter is indicated from your mention that changing the angle from 0 to 45 degrees leads to an anisotropic friction. It seems related to the transition from point to linear contact indicated in Fig. S6B but it is not stated clearly. Please detail this point.

(Remarks on code availability)

The code is a fairly fundamental ODE solution for simulating the oscillator force and displacement on leg setae. It supports the results shown in the videos.

Reviewer #2

(Remarks to the Author)

The authors present a work on small robots that move thanks to vibrational motion and elastic elements called setae that interact with the ground or walls of the environment. The authors describe the integration of some AI properties within these small prototypes.

- There seems not to be references on the use of what the authors call setae from the state-of-the-art. This is a pretty well known subject and the corpus of literature is abundant.

- There seems to be next to no methodological part that describe the mechanical behavior of the locomotion system. Some information can be extracted (I should say extrapolated...) from the Results section, but with no clear indication that the underlying theory has been understood.

- several typos, e.g. "combing" instead of "combining", "paradigm" instead of "paradigms", "by programmable actuating method" instead of "by a programmable actuating method", all in page 6. There are multiple typos in the table S1, as well. The rest of the paper generally has the same quality and attention to detail.

- poorly thought-out structure; the authors present a lot of data but in a very unorganized way. Much of the data is episodic, and no clear design of experiment is presented for the varied case studies that are shown.

Final remark: in my opinion the proposed article is not of sufficiently high quality to be considered for Nature Communications. It has neither a solid theoretical background (kinematics of the setae locomotion system is missing, dynamics is missing altogether), nor it has a particularly precise and comprehensive set of results to back the proposed concept - which by itself is not groundbreaking enough to justify the omissions.

(Remarks on code availability)

Reviewer #3

(Remarks to the Author)

This work present Flexible Electronic Robots (FEbots) with integrated programmable electronic modules. The authors have demonstrated various scenarios to show the maneuverability, agility and robustness of the FEbots. The movies show very interesting demonstrations to understand the performance of FEbots.

However, I have several points that I'm concerned as follows.

1. The FEbots are driven by the vibration of entire robot unit. The vibration frequency plays a very important role regarding the robot performance. But I can't find any related study, please add this part of study in the manuscript (e.g. Speed vs. frequency; Gait motion vs. frequency.)
2. The Speed vs. terrain data presents the robot speed on three different surfaces: sandpaper, A4 paper and smooth wood. Can you show each roughness quantitatively? Beside hard surface, I'm interested to know if the robot can move on sand or soil?
3. For the material of the seta, why SMA is chosen? Can other types of elastic materials be used?
4. With the integrated battery, how long the robot can move? What is the energy efficiency for the untethered robot?
5. In the movies, FEbots show the capabilities of moving forward, backward and turning. But it seems that the directional control is not very precise. Can you explain more in detail about this? How you can improve this performance?

(Remarks on code availability)

Version 1:

Reviewer comments:

Reviewer #1

(Remarks to the Author)

On review of the rebuttal, the concerns identified in the original draft have been suitably addressed in this revision. I believe that the paper is now publishable.

(Remarks on code availability)

The code is a fairly fundamental ODE solution for simulating the oscillator force and displacement on leg setae. It supports the results shown in the videos.

Reviewer #3

(Remarks to the Author)

The authors have addressed all my previous questions. I suggest to accept the the paper for publication in Nature Communication.

(Remarks on code availability)

Dear Reviewers,

Thank you for your constructive feedback on our manuscript titled "AI-embodied multi-modal flexible electronic robots with programmable sensing, actuating and self-learning." We have thoroughly addressed the reviewers' concerns through revisions to the text, figures, and supplementary materials. Below, we provide point-by-point responses to the reviewers' comments. All changes in the manuscript are highlighted in blue text.

Reviewer #1:

Comment #1-0:

This paper presents modular, flexible, centimetre scale robots called FEbots, inspired by millipedes and designed for embodied intelligence control. The use of vibrating SMA setae for multi-modal modular locomotion is of particular interest. Good performance on flat surfaces for this robot and interesting morphability through SMA actuation of setae and adaptation to different locomotion modes is also shown.

Reply #1-0:

We thank the reviewer 1 for the positive comments and helpful suggestions to help us improve the quality of our work. We have modified this manuscript according to the constructive comments.

Comment #1-1:

The standard of English in the paper is reasonable with good spelling and sentence structure, but still needs improvement. Please go over the paper carefully with a grammar checker or experienced writer of English and correct the occasional English grammatical inconsistencies. Examples of this that can be corrected in the abstract alone include definite articles "The FEbots proposed in this work"; "Inspired by the millipede" prepositions "(e.g. on smooth and rough ground, stairs, underwater, in vertical pipes)", indefinite articles "assisted by a modularization design method", and plural "in complex environments".

Reply #1-1:

We thank reviewer #1 for reviewing our manuscript carefully. We polish the English writing in the whole manuscript. For the detailed revisions, please see the revised manuscript.

Comment #1-2:

Fig. 1c caption 3 should read "Underwater crawling with waterproofing" or "Waterproof crawling underwater". Caption 6 "Climb up in slit with compressibility" should be more clearly worded - should this be "vertical pathway" as in Movie S4? Please keep consistent terminology across your materials.

Reply #1-2:

*We correct the figure according to the comment, please see new **Fig. 1c**.*

Fig. 1 (c) Schematic diagram illustrating FEbots interacting with the physical world via autonomous sensing-knowing-actuation.

Comment #1-3:

Fig. 1d is an important comparison to previous work, but the small size of this figure and the difficulty in discriminating the provided references by overlapping color regions on the radial graph makes it almost unreadable. This should be enlarged and the separate data more clearly

differentiated (perhaps line-only with different line styles?) on to provide a clearer fit with Table S1.

Reply #1-3:

We correct the figure according to the comment, please see new **Fig. 1d**.

Fig. 1 (d) Performance comparison between FEbots and reported small-scale soft robots.

Comment #1-4:

You mention that "The anisotropic friction between setae and ground is the key point for (the) locomotion mechanism" (again, please add definite article), but it is not clear how this anisotropic friction arises. Reference 34 mentions "bioinspired microrobotic legs with claws that are capable of inducing anisotropic friction" but it does not appear you are using such a design as movie S1 shows only a SMA wire. Are you relying on a squared-off corner on the setae to produce the difference between F_{f-} and F_{f+} or is it entirely or is it due to the $\theta=60$ angle of movement in which case it is more akin to a ratchet action? The latter is indicated from your mention that changing the angle from 0 to 45 degrees leads to an anisotropic friction. It seems related to the transition from point to linear contact indicated in Fig. S6B but it is not stated clearly. Please detail this point.

Reply #1-4:

We agree with this comment. The anisotropic friction relies on the tip of setae to produce the difference between F_{f-} and F_{f+} . The revision in the manuscript is as following:

“Anisotropic friction originates from backward-oriented seta (e.g. earthworm)^{23-26,34}, where angled setae tips generate direction-dependent resistance ($F_{f-} > F_{f+}$).”

“Speed reductions occur when seta lengths exceed 3, 7, 27, and 21 mm for $d = 0.07, 0.1, 0.2,$ and 0.3 mm, corresponding to transitions from point ($F_{f-} > F_{f+}$) to linear ($F_{f-} \approx F_{f+}$) contact between setae and substrate (**Fig. S7B**).”

“Heating SMA springs with 0.2 A current increases the foldable setae angle θ from 0° (linear contact, $F_{f-} \approx F_{f+}$) to 45° (point contact, $F_{f-} > F_{f+}$), inducing anisotropic substrate friction (**Fig. S11B, Movie S5**).”

22. H. Lu, M. Zhang, Y. Yang, Q. Huang, T. Fukuda, Z. Wang, Y. Shen, A bioinspired multilegged soft millirobot that functions in both dry and wet conditions. *Nat. Commun.* **9**, 3944 (2018).
23. L. Calabrese, A. Berardo, D. De Rossi, M. Gei, Nicola M. Pugno & G. Fantoni, A soft robot structure with limbless resonant, stick and slip locomotion. *Smart Mater. Struct.* **28**, 104005 (2019).
24. Y. Yan, L. Shui, S. Liu, Z. Liu, Y. Liu, Terrain adaptability and optimum contact stiffness of vibrobot with arrayed soft legs. *Soft robot.* **9**(5),981-990 (2022).
25. P. Lou, L. Tian, M. Yao, J. Nie, Y. He. Photothermal-driven crawlable soft robot with bionic earthworm-like bristles structure. *Adv. Intell. Syst.* **6**, 2300540 (2023).
26. L. V. Nguyen, K. T. Nguyen, and V. A. Ho, Terradynamics of monolithic soft robot driven by vibration mechanism. *IEEE Trans. Robot.* **41**, 1436-1455 (2025).
34. X. Yang, L. Chang & N. O. Pérez-Arancibia. An 88-milligram insect-scale autonomous crawling robot driven by a catalytic artificial muscle. *Sci. Robot.* **5**, eaba0015 (2020).

Reviewer #2:

Comment #2-0:

The authors present a work on small robots that move thanks to vibrational motion and elastic elements called setae that interact with the ground or walls of the environment. The authors describe the integration of some AI properties within these small prototypes.

Reply #2-0:

We thank the reviewer 2 for the helpful suggestions to help us improve the quality of our work. We have modified the manuscript according to the constructive comments.

Comment #2-1:

There seems not to be references on the use of what the authors call setae from the state-of-the-art. This is a pretty well known subject and the corpus of literature is abundant.

Reply #2-1:

We have added several references about research on setae in the revised manuscript.

“Anisotropic friction originates from backward-oriented setae (e.g. earthworm)^{23-26,34}, where angled setae tips generate direction-dependent resistance ($F_{f-} > F_{f+}$).”

22. H. Lu, M. Zhang, Y. Yang, Q. Huang, T. Fukuda, Z. Wang, Y. Shen, A bioinspired multilegged soft millirobot that functions in both dry and wet conditions. *Nat. Commun.* **9**, 3944 (2018).
23. L. Calabrese, A. Berardo, D. De Rossi, M. Gei, Nicola M. Pugno & G. Fantoni, A soft robot structure with limbless resonant, stick and slip locomotion. *Smart Mater. Struct.* **28**, 104005 (2019).
24. Y. Yan, L. Shui, S. Liu, Z. Liu, Y. Liu, Terrain adaptability and optimum contact stiffness of vibrobot with arrayed soft legs. *Soft robot.* **9**(5),981-990 (2022).
25. P. Lou, L. Tian, M. Yao, J. Nie, Y. He. Photothermal-driven crawlable soft robot with bionic earthworm-like bristles structure. *Adv. Intell. Syst.* **6**, 2300540 (2023).
26. L. V. Nguyen, K. T. Nguyen, and V. A. Ho, Terradynamics of monolithic soft robot driven by vibration mechanism. *IEEE Trans. Robot.* **41**, 1436-1455 (2025).
34. X. Yang, L. Chang & N. O. Pérez-Arancibia. An 88-milligram insect-scale autonomous crawling robot driven by a catalytic artificial muscle. *Sci. Robot.* **5**, eaba0015 (2020).

Comment #2-2:

There seems to be next to no methodological part that describe the mechanical behavior of the locomotion system. Some information can be extracted (I should say extrapolated...) from the Results section, but with no clear indication that the underlying theory has been understood.

Reply #2-2:

*We appreciate Reviewer #2 for the important evaluation. We conducted a deeper analysis of the robot's dynamic model. By employing "Cosserat elastic rod theory", we establish the dynamic model for setae deformation, while combining spring-damper theory to analyze the reciprocal sliding behavior of the robot in the X-direction. This approach validates the accuracy of the FEbot's dynamic model. Please check the details in **Move S1** and **Section 2** in the supplementary materials.*

Note S2: Simulation and experimental results for the detailed locomotion of FEbot

1. Dynamic models

The dynamic model of the FEbot is derived as follows (**Fig. S3a**). Assuming the motor's force is uniformly transmitted to each seta, the system is modeled as a classical spring-damper system, a standard approach for vibratory robotic motion. The robot's mass is lumped at the geometric center C_G of the driver, located at coordinates (x, y) . The governing equations are:

$$\begin{cases} F_x - F_{f+} - \sum_i^4 (k'_i x_b + v'_i \dot{x}_b) = m_F \ddot{x} & \dot{x} > 0 \\ F_x - F_{f-} - \sum_i^4 (k'_i x_b + v'_i \dot{x}_b) = m_F \ddot{x} & \dot{x} \leq 0 \end{cases} \quad i = 1,2,3,4 \quad (\text{S2})$$

$$\begin{cases} F_y + mg - F_N = m_F \ddot{y} \\ F_N = \sum_i^4 F_{ni} \\ F_{ni} = k_i y + v_i \dot{y} \end{cases} \quad i = 1,2,3,4 \quad (\text{S3})$$

where F_x and F_y are the oscillator's input forces in the X- and Y-directions, respectively; m is the FEbot's mass; g is gravitational acceleration; x and y are displacements in the X- and Y-directions; F_{f+} and F_{f-} denote the friction forces opposing forward and backward sliding motions, respectively; k'_i and v'_i are the stiffness and damping coefficients in the X-direction for the i -th seta; F_{ni} is the normal contact force of the i -th seta; F_N is the total normal contact force between the FEbot and substrate; k_i and v_i are the stiffness and

damping coefficients in the Y-direction; x_b and \dot{x}_b represent the displacement and velocity of the seta tip relative to point O_1 in the X_1 -direction of the $X_1O_1Y_1$ coordinate system.

As shown in Fig. **S3(b)**, x_b and \dot{x}_b can be obtained by using Cosserat rod theory which is used to model the bending shape of each seta [47].

$$\left\{ \begin{array}{l} \mathbf{p}_s = \mathbf{R}\mathbf{v} \\ \mathbf{p}_t = \mathbf{R}\mathbf{q} \\ \mathbf{R}_s = \mathbf{R}\hat{\mathbf{u}} \\ \mathbf{R}_t = \mathbf{R}\hat{\boldsymbol{\omega}} \\ \mathbf{n}_s = \rho A \mathbf{R}(\hat{\boldsymbol{\omega}}\mathbf{q} + \mathbf{q}_t) - \mathbf{f} \\ \mathbf{m}_s = \partial_t(\mathbf{R}\rho\mathbf{J}\boldsymbol{\omega}) - \hat{\mathbf{p}}_s\mathbf{n} - \mathbf{l} \\ \mathbf{q}_s = \mathbf{v}_t - \hat{\mathbf{u}}\mathbf{q} + \hat{\boldsymbol{\omega}}\mathbf{v} \\ \boldsymbol{\omega}_s = \mathbf{u}_t - \hat{\mathbf{u}}\boldsymbol{\omega} \end{array} \right. \quad (\text{S4})$$

where \mathbf{p} denotes the global position; \mathbf{R} represents the rotation matrix; \mathbf{v} is the rate of change of position with respect to arclength in the local frame; \mathbf{q} and $\boldsymbol{\omega}$ are the linear velocity and angular velocity in the local coordinate system, respectively; \mathbf{u} is the curvature vector in the local frame; \mathbf{n} is the internal force; \mathbf{m} is the internal moment; $\hat{\cdot}$ indicates the conversion of the variables into a skew-symmetric matrix. \mathbf{f} is the distributed force in the global frame; ρ is material density; A is the cross-sectional area; \mathbf{J} is the second mass moment of inertia tensor.

The moment of inertia can be expanded as:

$$\partial_t(\mathbf{R}\rho\mathbf{J}\boldsymbol{\omega}) = \rho\mathbf{R}(\hat{\boldsymbol{\omega}}\mathbf{J}\boldsymbol{\omega} + \mathbf{J}\boldsymbol{\omega}_t) \quad (\text{S5})$$

Considering the self-weight of the bristle and the aerodynamic resistance, the distributed force \mathbf{f} can be expressed as:

$$\mathbf{f} = \rho A \mathbf{g} - \mathbf{R}\mathbf{C}\mathbf{q} \odot |\mathbf{q}| + \bar{\mathbf{f}} \quad (\text{S6})$$

where $\bar{\mathbf{f}}$ encompasses any residual forces not explicitly modeled, and the symbol \odot denotes the Hadamard product (element-wise multiplication):

$$\mathbf{q} \odot |\mathbf{q}| = [q_1^2 \text{sgn}(q_1) \quad q_2^2 \text{sgn}(q_2) \quad q_3^2 \text{sgn}(q_3)]^T \quad (\text{S7})$$

From the material constitutive law, we have:

$$\begin{cases} \mathbf{n} = \mathbf{R}[\mathbf{K}_{se}(\mathbf{v} - \mathbf{v}^*) + \mathbf{B}_{se}\mathbf{v}_t] \\ \mathbf{m} = \mathbf{R}[\mathbf{K}_{bt}(\mathbf{u} - \mathbf{u}^*) + \mathbf{B}_{bt}\mathbf{u}_t] \end{cases} \quad (\text{S8})$$

Fig. S3. Simplified dynamic model of the FEbot. (a) Spring-damper system representation. (b) Seta force analysis based on Cosserat theory.

For the partial differential equation (PDE) system (4), a specific implicit differentiation scheme can be applied to discretize the time derivative terms, thereby reducing the problem to a spatial ordinary differential equation (ODE). For discretized variables, the superscript notation is used to denote the time step index. A general first-order implicit differentiation formula can be uniformly written in the following form:

$${}^{(i)}y_t \approx c_0 {}^{(i)}y + \sum_{k=1}^{\infty} [c_k {}^{(i-k)}y + d_k {}^{(i-k)}y_t] = c_0 {}^{(i)}y + {}^{(i)h}y \quad (\text{S9})$$

Among them, ${}^{(i)h}y$ represents the sum of all remaining terms dependent on the historical values of y . In this paper, the backward differentiation formulas (BDF)- α method is employed to solve the system, which can be expressed as:

$${}^{(i)}y_t = c_0 {}^{(i)}y + c_1 {}^{(i-1)}y + c_2 {}^{(i-2)}y + d_1 {}^{(i-1)}y_t \quad (\text{S10})$$

where:

$$\begin{cases} c_0 = (1.5 + \alpha)/[\delta t(1 + \varrho)] \\ c_1 = -2/\delta t \\ c_2 = (0.5 + \varrho)/[\delta t(1 + \varrho)] \\ d_1 = \alpha/(1 + \varrho) \end{cases} \quad (\text{S11})$$

Using this method, after discretizing the time derivatives, the distributed forces and moments of inertia are substituted into Eq. (S4). The partial differential equations are thereby simplified into a system of ordinary differential equations with respect to arc length:

$$\begin{cases} \mathbf{p}_s = \mathbf{R}\mathbf{v} \\ \mathbf{R}_s = \mathbf{R}\hat{\mathbf{u}} \\ \mathbf{n}_s = \mathbf{R}[\rho A(\hat{\boldsymbol{\omega}}\mathbf{q} + \mathbf{q}_t) + \mathbf{C}\mathbf{q} \odot |\mathbf{q}|] - \rho A\mathbf{g} - \bar{\mathbf{f}} \\ \mathbf{m}_s = \rho\mathbf{R}(\hat{\boldsymbol{\omega}}\mathbf{J}\boldsymbol{\omega} + \mathbf{J}\boldsymbol{\omega}_t) - \hat{\mathbf{p}}_s\mathbf{n} - \mathbf{l} \\ \mathbf{q}_s = \mathbf{v}_t - \hat{\mathbf{u}}\mathbf{q} + \hat{\boldsymbol{\omega}}\mathbf{v} \\ \boldsymbol{\omega}_s = \mathbf{u}_t - \hat{\mathbf{u}}\boldsymbol{\omega} \end{cases} \quad (\text{S12})$$

The linear constitutive relationship can be written as:

$$\begin{cases} \mathbf{v} = (\mathbf{K}_{se} + c_0\mathbf{B}_{se})^{-1}(\mathbf{R}^T\mathbf{n} + \mathbf{K}_{se}\mathbf{v}^* - \mathbf{B}_{se}^h\mathbf{v}) \\ \mathbf{u} = (\mathbf{K}_{bt} + c_0\mathbf{B}_{bt})^{-1}(\mathbf{R}^T\mathbf{m} + \mathbf{K}_{bt}\mathbf{u}^* - \mathbf{B}_{bt}^h\mathbf{u}) \end{cases} \quad (\text{S13})$$

All time derivative terms in Eq. (S13) are computed from the state variables at the current or previous time steps using the following equation:

$$\begin{cases} \mathbf{v}_t = c_0\mathbf{v} + \mathbf{v}^h \\ \mathbf{u}_t = c_0\mathbf{u} + \mathbf{u}^h \\ \mathbf{q}_t = c_0\mathbf{q} + \mathbf{q}^h \\ \boldsymbol{\omega}_t = c_0\boldsymbol{\omega} + \boldsymbol{\omega}^h \end{cases} \quad (\text{S14})$$

By applying the boundary condition of the force $\mathbf{n}_L(t) = [F_{ni}(t), F_{fi}(t)]$, $\mathbf{m}_L(t) = 0$ acting on the bristle tip and solving the ODE system in Eqs. (12) to (14), x_b and \dot{x}_b can be obtained according to the following equation:

$$\begin{cases} \mathbf{p}_x = f(\mathbf{p}_y) = 0.7592\mathbf{p}_y + 0.0488 \\ \mathbf{q}_x = g(\mathbf{q}_y) = -0.7379\mathbf{q}_y \\ \mathbf{p}_x = x_b, \mathbf{p}_y = y, \mathbf{q}_x = \dot{x}_b, \mathbf{q}_y = \dot{y} \end{cases} \quad (\text{S15})$$

Finally, the displacement x and y in X and Y direction can be obtained, respectively.

2. Solution approaches to the dynamical models

According to the experimental results shown in **Fig. S4a**, the input force for the simulation can be expressed by:

$$F_x = 0.142 \sin(1632t) \quad (\text{S16})$$

$$F_y = 0.097 \sin(1632(t - 0.0014)) + 0.002079 \quad (\text{S17})$$

To accurately model the locomotion, the force in Y direction needs to be modified as:

$$F_y = \gamma \{0.097 \sin1632(t - 0.0014) + 0.002079\} \quad (\text{S18})$$

where $\gamma=1.4$ is the coefficient of correction.

The FEbot demonstrates vibration in Y direction when moving forward. The stiffness of setae in Y direction needs to be measured as well. As shown in **Fig. S4b**, the stiffness k_i is expressed by:

$$k_i = \frac{Pg}{4(H-H_1)}, \quad i = 1,2,3,4 \quad (\text{S19})$$

where P is the applied load; H and H_1 are the height of the FEbot with and without load, respectively.

The normal contact force of each seta F_{ni} according to Eq. (S3, S18) can be used to calculate x_b and \dot{x}_b . To simplify the simulation method, the friction between substrate and each seta F_{fi} is expressed as:

$$F_{fi} = \gamma_2 F_{ni}. \quad (\text{S20})$$

where $\gamma_2=0.18$ is the coefficient of correction.

Fig. S4. Input force and stiffness identification. (a) Simulated input force calibrated against experimental data (Fig. S2c). (b) Seta stiffness measurement: average stiffness = 24.5 N/m (individual measurements: 2.5, 2.63, and 2.38 N/mm).

Fig. S5. Seta deformation based on the Cosserat model. (a) Time-resolved bending process. **(b)** Tip displacement (x_b in X_1 direction and y in Y_1 direction). **(c)** Tip velocity (\dot{x}_b in X_1 direction and \dot{y} in Y_1 direction). **(d)** Five-cycle Y-direction displacement: simulation vs. high-speed camera data **Movie S1**). **(e)** High-speed camera experimental setup.

Figure S5a illustrates the bending process of the seta over time. **Fig. S5b-c** present the simulation results of the tip's displacement (x_b in X_1 direction and y in Y_1 direction) and speed (\dot{x}_b in X_1 direction and \dot{y} in Y_1 direction) relative to point O1 in the $X_1O_1Y_1$ coordinate system, respectively. The Y_1 -direction displacement in **Fig. S5B** is transformed into the vertical amplitude in the global Y-direction, which agrees qualitatively with experimental observations. Thus, x_b and \dot{x}_b are assumed to be accurately obtained and utilized in Eq. (S2) to calculate the displacement x in the X-direction.

The friction in Eq. (S2) can be expressed by:

$$\begin{cases} F_{f-} = \mu_- F_N \\ F_{f+} = \mu_+ F_N \end{cases} \quad (\text{S21})$$

where μ_- and μ_+ are positive friction coefficients satisfying $\mu_- \gg \mu_+$. These coefficients $\mu(\mu_-, \mu_+)$ are measured via the method depicted in **Fig. S3**. The FEbot is placed on an inclined surface, and the sliding velocity varies with the inclination angle β . Under uniform sliding motion, the equilibrium equations yield:

$$\begin{cases} mg \sin \beta = \mu F_N \\ F_N = mg \cos \beta \end{cases} \quad (\text{S22})$$

Then, friction coefficient $\mu(\mu_-, \mu_+) = \tan \beta = \tan(\arctan(\frac{h}{\sqrt{l^2 - h^2}}))$. l and h are the length and height of inclined surface. These coefficients $\mu(\mu_-, \mu_+)$ are measured via the method depicted in **Fig. S6a**. **Fig. S6b** shows the steady-state displacement of the seta's contact point in the X-direction, validating the motion principles. Additional simulation data and parameters are provided in **Fig. S6c-e** and **Table S3**.

Fig. S6. Motion simulation results. (a) Friction coefficient measurement methodology. **(b)** Five-cycle X-direction displacement: simulation vs. experiment (**Movie S1**). **(c)** Forward motion friction analysis. **(d)** X-direction speed profile. **(e)** Y-direction speed profile.

Table S3. Simulation Parameters Based on Cosserat Theory

k_i (N/m)	24.5/4	μ_+	0.5773
ν_i	0.0316011/4	μ_-	0.7536
m (kg)	0.83×10^{-3}	γ	1.4
g (N/kg)	9.8	L (m)	0.005
k'_i	0.006/4	E (Pa)	7.3×10^{10}
ν'_i	0.03/4	G (Pa)	2.81×10^{10}
d (mm)	0.1	ρ (g/cm ³)	6.5
θ	45°	A (m ²)	7.854×10^{-9}
J (m ⁴)	$\begin{bmatrix} 4.91e^{-18} & 0 & 0 \\ 0 & 4.91e^{-18} & 0 \\ 0 & 0 & 9.82e^{-18} \end{bmatrix}$	K_{se} (N)	$\begin{bmatrix} 220.5 & 0 & 0 \\ 0 & 220.5 & 0 \\ 0 & 0 & 573.3 \end{bmatrix}$
K_{bt} (Nm ²)	$\begin{bmatrix} 3.58e^{-7} & 0 & 0 \\ 0 & 3.58e^{-7} & 0 \\ 0 & 0 & 2.76e^{-7} \end{bmatrix}$	B_{se} (Ns)	$0_{3 \times 3}$
C	$0_{3 \times 3}$	B_{bt}	$0_{3 \times 3}$
ϱ	-0.48	c_1	-6.51×10^3
c_0	6.39×10^3	c_2	125.3
c_1	-6.51×10^3	d_1	-0.9231

Comment #2-3:

several typos, e.g. "combing" instead of "combining", "paradigm" instead of "paradigms", "by programmable actuating method" instead of "by a programmable actuating method", all in page 6. There are multiple typos in the table S1, as well. The rest of the paper generally has the same quality and attention to detail.

Reply #2-3:

We thank reviewer #2 for reviewing our manuscript carefully. We polish the English writing in the whole manuscript. For the detailed revisions, please see the revised manuscript and supplementary materials.

Comment #2-4:

poorly thought-out structure; the authors present a lot of data but in a very unorganized way. Much of the data is episodic, and no clear design of experiment is presented for the varied case studies that are shown.

Reply #2-4:

*The manuscript is systematically structured to progressively validate the design and intelligence of the embodied AI robot: Beginning with the overarching concept of multimodal perception and locomotion in **Fig. 1**, the study transitions to rigorously establishing the dynamic model and gait analysis under variable structural parameters in **Fig. 2**. This theoretical foundation is then experimentally extended in **Fig. 3** and **Fig. 4**, which quantitatively demonstrate the robot's multimodal locomotion and perception capabilities, directly aligning with the conceptual framework. Finally, **Fig. 5** culminates the design philosophy by implementing embodied intelligence (autonomous navigation, environmental mapping) via hyperdimensional computing, thereby creating a closed-loop validation from theory to functional intelligence. Each figure incrementally addresses specific hypotheses, ensuring methodological coherence across all case studies.*

*We also have rewritten the manuscript and reorganized some figures, such as **Fig. 1**, **Fig. 2**, **Fig. 3**, **Fig. S3 to S6**, **Movie S10-S12**, **Table S1** and **Table S3**.*

Comment #2-5:

Final remark: in my opinion the proposed article is not of sufficiently high quality to be considered for Nature Communications. It has neither a solid theoretical background (kinematics of the setae locomotion system is missing, dynamics is missing altogether), nor it has a particularly precise and comprehensive set of results to back the proposed concept - which by itself is not groundbreaking enough to justify the omissions.

Reply #2-5:

*To improve the quality of this work, we conduct a deeper analysis of the robot's dynamic model. By employing "Cosserat elastic rod theory", we establish the dynamic model for setae deformation, while combining spring-damper theory to analyze the reciprocal sliding behavior of the robot in the X-direction. This approach validates the accuracy of the FEbot's dynamic model. Please check the details in **Movie S1** and **Note S2** in the supplementary materials. Furthermore, we rewrite the manuscript and reorganize some figures to better explain the experimental results, according to your comment. Therefore, we strongly believe that this work is qualified to be published in Nature Communication after our careful revision.*

Reviewer #3:

Comment #3-0:

This work present Flexible Electronic Robots (FEbots) with integrated programmable electronic modules. The authors have demonstrated various scenarios to show the maneuverability, agility and robustness of the FEbots. The movies show very interesting demonstrations to understand the performance of FEbots.

Reply #3-0:

We appreciate Reviewer #3 for the positive evaluation. As all the comments from Review #3 are important, we have revised the manuscript based on the comments and suggestions.

Comment #3-1:

The FEbots are driven by the vibration of entire robot unit. The vibration frequency plays a very important role regarding the robot performance. But I can't find any related study, please add this part of study in the manuscript (e.g. Speed vs. frequency; Gait motion vs. frequency.)

Reply #3-1:

*We have provided the details in the revised manuscript (**Fig. 2e**).*

Fig. 2 (e) Forward speed as a function of driving frequency for a unit with $L = 5$ mm, $d = 0.1$ mm, and $\theta = 45^\circ$.

Comment #3-2:

The Speed vs. terrain data presents the robot speed on three different surfaces: sandpaper, A4 paper and smooth wood. Can you show each roughness quantitatively? Beside hard surface, I'm interested to know if the robot can move on sand or soil?

Reply #3-2:

We agree with this comment. We add the roughness of sandpaper, A4 paper and smooth wood in the revised manuscript:

“Motion performance varies across terrains: smooth wood ($R_a = 0.5 \mu\text{m}$), A4 paper ($R_a = 2 \mu\text{m}$), and sandpaper ($R_a = 8 \mu\text{m}$) (**Fig. 3f**).”

In conclusion section, we add “Current challenges and future research directions focus on: (i) although the robots exhibit strong mobility, they cannot crawl on soft terrain like sand or soil, due to the setae sinking into loose surfaces, rendering the robot immobile.”

Comment #3-3:

For the material of the seta, why SMA is chosen? Can other types of elastic materials be used?

Reply #3-3:

We have provided the details in the revised manuscript. “Their locomotion employs oscillatory actuation driven by periodic bending of super-elastic alloy (SSMA) setae. SSMA provides elastic deformation capacity, corrosion resistance, and durability (**Table S1**), enabling integration with flexible electronics.”

*We have provided the details in the revised supplementary materials (**Table S1**), and the “Seta Materials Comparison” is provided.*

Table S1 Seta Materials Comparison

Reference	Materials	Mechanical Properties	Durability	Environmental Adaptability
[22], [24-26],	Polydimethylsiloxane (PDMS)/Silicone	Advantages: Low elastic modulus (0.1-5 MPa), high softness and deformability Limitations: Prone to plastic deformation under large strains (e.g., permanent compression in silicone)	Limitations: UV/ozone-induced aging	Limitations: Thermal degradation (silicone softens >200°C, PDMS decomposes >300°C)
[23]	Resin	Advantages: Tunable elastic modulus (1-10 GPa), photocurable resins enable precise complex structures. Limitations: High brittleness, susceptibility to cracking	Limitations: Low stress fatigue threshold (cracking at <math>10^4</math> cycles)	Limitations: High thermal expansion coefficient (50-100 ppm/°C), solvent-induced swelling
This work	SSMA	Advantages: Exceptional elastic deformation capacity, high elastic modulus (~30-80 GPa), suitable for high-load applications Limitations: High rigidity	Advantages: Long fatigue life (10^6-10^7 cycles), wear resistance, no aging issues. Limitations: Phase transformation accumulation under cyclic loads	Advantages: Broad temperature tolerance (-200°C to 400°C), corrosion resistance

Comment #3-4:

With the integrated battery, how long the robot can move? What is the energy efficiency for the untethered robot?

Reply #3-3:

We agree with this comment. We add the detailed information in the revised manuscript:

“A FEbot configuration featuring bilateral setae maintains step-climbing capability following inversion (**Fig. 3i, Movie S12**). The system demonstrates 25 minutes of continuous operation with 2.5 Wh power consumption.”

Comment #3-5:

In the movies, FEbots show the capabilities of moving forward, backward and turning. But it seems that the directional control is not very precise. Can you explain more in detail about this? How you can improve this performance?

Reply #3-5:

We add the details about the insufficient directional motion control in the revised manuscript:

“The insufficient directional motion control accuracy of FEbots primarily stems from following mechanical characteristic variations: 1) friction coefficient discrepancy: manufacturing tolerances and assembly precision limitations lead to the differences in friction coefficients between setae modules and contact surfaces; 2) directional deviation accumulation: the speed mismatch between symmetrically distributed actuation modules induces cumulative directional errors during motion. To enhance control precision, a sensor-based closed-loop control scheme is proposed in the following sections: Implement optical encoders or attitude sensors to establish a pose perception system, which can achieve closed-loop regulation of turning angles through real-time feedback. This approach enables continuous trajectory correction, effectively suppressing directional drift through electromechanical coupling control. Different types of programmatically assembled FEbots are listed in **Table S4**.”